# OnACID: Online Analysis of Calcium Imaging Data in Real Time

**Andrea Giovannucci**†[1]    **Johannes Friedrich**†*[1]    **Matthew Kaufman**‡

**Anne K. Churchland**‡    **Dmitri Chklovskii**†    **Liam Paninski***

**Eftychios A. Pnevmatikakis**†[2]

† Flatiron Institute, New York, NY 10010
‡ Cold Spring Harbor Laboratory, Cold Spring Harbor, NY 11724
* Columbia University, New York, NY 10027
{agiovannucci, jfriedrich, dchklovskii, epnevmatikakis}@flatironinstitute.org
{mkaufman, churchland}@cshl.edu
liam@stat.columbia.edu

## Abstract

Optical imaging methods using calcium indicators are critical for monitoring the activity of large neuronal populations in vivo. Imaging experiments typically generate a large amount of data that needs to be processed to extract the activity of the imaged neuronal sources. While deriving such processing algorithms is an active area of research, most existing methods require the processing of large amounts of data at a time, rendering them vulnerable to the volume of the recorded data, and preventing real-time experimental interrogation. Here we introduce OnACID, an Online framework for the Analysis of streaming Calcium Imaging Data, including i) motion artifact correction, ii) neuronal source extraction, and iii) activity denoising and deconvolution. Our approach combines and extends previous work on online dictionary learning and calcium imaging data analysis, to deliver an automated pipeline that can discover and track the activity of hundreds of cells in real time, thereby enabling new types of closed-loop experiments. We apply our algorithm on two large scale experimental datasets, benchmark its performance on manually annotated data, and show that it outperforms a popular offline approach.

## 1 Introduction

Calcium imaging methods continue to gain traction among experimental neuroscientists due to their capability of monitoring large targeted neuronal populations across multiple days or weeks with decisecond temporal and single-neuron spatial resolution. To infer the neural population activity from the raw imaging data, an analysis pipeline is employed which typically involves solving the following problems (all of which are still areas of active research): i) correcting for motion artifacts during the imaging experiment, ii) identifying/extracting the sources (neurons and axonal or dendritic processes) in the imaged field of view (FOV), and iii) denoising and deconvolving the neural activity from the dynamics of the expressed calcium indicator.

The fine spatiotemporal resolution of calcium imaging comes at a data rate cost; a typical two-photon (2p) experiment on a 512×512 pixel large FOV imaged at 30Hz, generates ∼50GB of data (in 16-bit integer format) per hour. These rates can be significantly higher for other planar and volumetric imaging techniques, e.g., light-sheet [1] or SCAPE imaging [4], where the data rates can exceed 1TB per hour. The resulting data deluge poses a significant challenge.

Of the three basic pre-processing problems described above, the problem of source extraction faces the most severe scalability issues. Popular approaches reshape the data movies into a large array with dimensions (#pixels) × (#timesteps), that is then factorized (e.g., via independent component analysis [20] or constrained non-negative matrix factorization (CNMF) [26]) to produce the locations in the FOV and temporal activities of the imaged sources. While effective for small or medium datasets, direct factorization can be impractical, since a typical experiment can quickly produce datasets larger than the available RAM. Several strategies have been proposed to enhance scalability, including parallel processing [9], spatiotemporal decimation [10], dimensionality reduction [23], and out-of-core processing [13]. While these approaches enable efficient processing of larger datasets, they still require significant storage, power, time, and memory resources.

Apart from recording large neural populations, optical methods can also be used for stimulation [5]. Combining optogenetic methods for recording and perturbing neural ensembles opens the door to exciting closed-loop experiments [24, 15, 8], where the pattern of the stimulation can be determined based on the recorded activity during behavior. In a typical closed-loop experiment, the monitored/perturbed regions of interest (ROIs) have been preselected by analyzing offline a previous dataset from the same FOV. Monitoring the activity of a ROI, which usually corresponds to a soma, typically entails averaging the fluorescence over the corresponding ROI, resulting in a signal that is only a proxy for the actual neural activity and which can be sensitive to motion artifacts and drifts, as well as spatially overlapping sources, background/neuropil contamination, and noise. Furthermore, by preselecting the ROIs, the experimenter is unable to detect and incorporate new sources that become active later during the experiment, which prevents the execution of truly closed-loop experiments.

In this paper, we present an Online, single-pass, algorithmic framework for the Analysis of Calcium Imaging Data (OnACID). Our framework is highly scalable with minimal memory requirements, as it processes the data in a streaming fashion one frame at a time, while keeping in memory a set of low dimensional sufficient statistics and a small minibatch of the last data frames. Every frame is processed in four sequential steps: i) The frame is registered against the previous denoised (and registered) frame to correct for motion artifacts. ii) The fluorescence activity of the already detected sources is tracked. iii) Newly appearing neurons and processes are detected and incorporated to the set of existing sources. iv) The fluorescence trace of each source is denoised and deconvolved to provide an estimate of the underlying spiking activity.

Our algorithm integrates and extends the online NMF algorithm of [19], the CNMF source extraction algorithm of [26], and the near-online deconvolution algorithm of [11], to provide a framework capable of real time identification and processing of hundreds of neurons in a typical 2p experiment (512×512 pixel wide FOV imaged at 30Hz), enabling novel designs of closed-loop experiments.

We apply OnACID to two large-scale (50 and 65 minute long) mouse *in vivo* 2p datasets; our algorithm can find and track hundreds of neurons faster than real-time, and outperforms the CNMF algorithm of [26] benchmarked on multiple manual annotations using a precision-recall framework.

## 2   Methods

We illustrate OnACID in process in Fig. 1. At the beginning of the experiment (Fig. 1-left), only a few components are active, as shown in the panel A by the max-correlation image[3], and these are detected by the algorithm (Fig. 1B). As the experiment proceeds more neurons activate and are subsequently detected by OnACID (Fig. 1 middle, right) which also tracks their activity across time (Fig. 1C). See also Supplementary Movie 1 for an example in simulated data.

Next, we present the steps of OnACID in more detail.

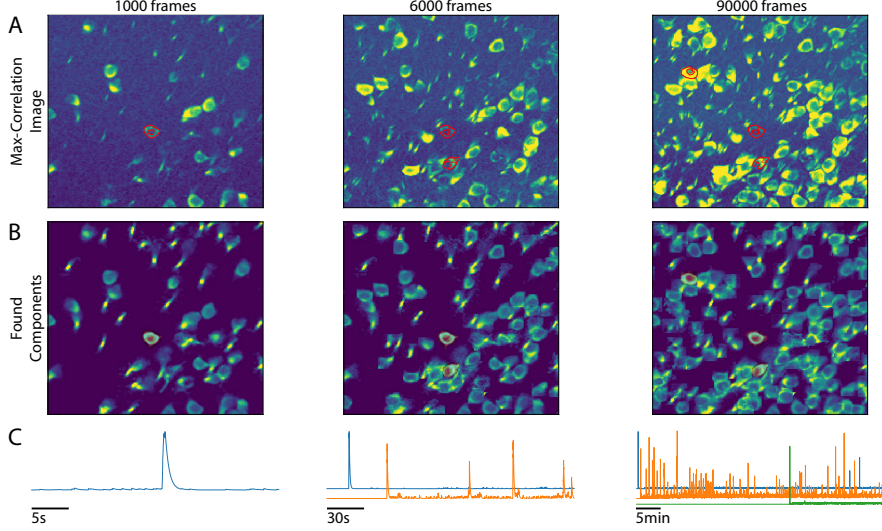

Figure 1: **Illustration of the online data analysis process.** Snapshots of the online analysis after processing 1000 frames (left), 6000 frames (middle), and 90000 frames (right). A) "Max-correlation" image of registered data at each snapshot point (see text for definition). B) Spatial footprints (shapes) of the components (neurons and processes) found by OnACID up to each point. C) Examples of neuron activity traces (marked by contours in panel A and highlighted in red in panel B). As the experiment proceeds, OnACID detects newly active neurons and tracks their activity.

**Motion correction:** Our online approach allows us to employ a very simple yet effective motion correction scheme: each *denoised* dataframe can be used to register the next incoming noisy dataframe. To enhance robustness we use the denoised background/neuropil signal (defined in the next section) as a template to align the next dataframe. We use rigid, sub-pixel registration [16], although piecewise rigid registration can also be used at an additional computational cost. This simple alignment process is not suitable for offline algorithms due to noise in the raw data, leading to the development of various algorithms based on template matching [14, 23, 25] or Hidden Markov Models [7, 18].

**Source extraction:** A standard approach for source extraction is to model the fluorescence within a matrix factorization framework [20, 26]. Let $Y \in \mathbb{R}^{d \times T}$ denote the observed fluorescence across space and time in a matrix format, where $d$ denotes the number of imaged pixels, and $T$ the length of the experiment in timepoints. If the number of imaged sources is $K$, then let $A \in \mathbb{R}^{d \times K}$ denote the matrix where column $i$ encodes the "spatial footprint" of the source $i$. Similarly, let $C \in \mathbb{R}^{K \times T}$ denote the matrix where each row encodes the temporal activity of the corresponding source. The observed data matrix can then be expressed as

$$Y = AC + B + E, \tag{1}$$

where $B, E \in \mathbb{R}^{d \times T}$ denote matrices for background/neuropil activity and observation noise, respectively. A common approach, introduced in [26], is to express the background matrix $B$ as a low rank matrix, i.e., $B = \mathbf{bf}$, where $\mathbf{b} \in \mathbb{R}^{d \times n_b}$ and $\mathbf{f} \in \mathbb{R}^{n_b \times T}$ denote the spatial and temporal components of the low rank background signal, and $n_b$ is a small integer, e.g., $n_b = 1, 2$. The CNMF framework of [26] operates by alternating optimization of $[A, \mathbf{b}]$ given the data $Y$ and estimates of $[\mathbf{C}; \mathbf{f}]$, and vice versa, where each column of $A$ is constrained to be zero outside of a neighborhood around its previous estimate. This strategy exploits the spatial locality of each neuron to reduce the computational complexity. This framework can be adapted to a data streaming setup using the online NMF algorithm of [19], where the observed fluorescence at time $t$ can be written as

$$\mathbf{y}_t = A\mathbf{c}_t + \mathbf{bf}_t + \varepsilon_t. \tag{2}$$

Proceeding in a similar alternating way, the activity of all neurons at time $t$, $\mathbf{c}_t$, and temporal background $\mathbf{f}_t$, given $\mathbf{y}_t$ and the spatial footprints and background $[A, \mathbf{b}]$, can be found by solving a nonnegative least squares problem, whereas $[A, \mathbf{b}]$ can be estimated efficiently as in [19] by only keeping in memory the sufficient statistics (where we define $\tilde{\mathbf{c}}_t = [\mathbf{c}_t; \mathbf{f}_t]$)

$$W_t = \tfrac{t-1}{t} W_{t-1} + \tfrac{1}{t} \mathbf{y}_t \tilde{\mathbf{c}}_t^\top, \qquad M_t = \tfrac{t-1}{t} M_{t-1} + \tfrac{1}{t} \tilde{\mathbf{c}}_t \tilde{\mathbf{c}}_t^\top, \tag{3}$$

while at the same time enforcing the same spatial locality constraints as in the CNMF framework.

**Deconvolution:** The online framework presented above estimates the demixed fluorescence traces $\mathbf{c}^1, \ldots, \mathbf{c}^K$ of each neuronal source. The fluorescence is a filtered version of the underlying neural activity that we want to infer. To further denoise and deconvolve the neural activity from the dynamics of the indicator we use the OASIS algorithm [11] that implements the popular spike deconvolution algorithm of [30] in a nearly online fashion by adapting the highly efficient Pool Adjacent Violators Algorithm used in isotonic regression [3]. The calcium dynamics is modeled with a stable autoregressive process of order $p$, $c_t = \sum_{i=1}^{p} \gamma_i c_{t-i} + s_t$. We use $p = 1$ here, but can extend to $p = 2$ to incorporate the indicator rise time [11]. OASIS solves a modified LASSO problem

$$\underset{\hat{\mathbf{c}}, \hat{\mathbf{s}}}{\text{minimize}} \quad \tfrac{1}{2}\|\hat{\mathbf{c}} - \mathbf{y}\|^2 + \lambda\|\hat{\mathbf{s}}\|_1 \quad \text{subject to} \quad \hat{s}_t = \hat{c}_t - \gamma\hat{c}_{t-1} \geq s_{\min} \text{ or } \hat{s}_t = 0 \tag{4}$$

where the $\ell_1$ penalty on $\hat{s}$ or the minimal spike size $s_{\min}$ can be used to enforce sparsity of the neural activity. The algorithm progresses through each time series sequentially from beginning to end and backtracks only to the most recent spike. We can further restrict the lag to few frames, to obtain a good approximate solution applicable for real-time experiments.

**Detecting new components:** The approach explained above enables tracking the activity of a fixed number of sources, and will ignore neurons that become active later in the experiment. To account for a variable number of sources in an online NMF setting, [12] proposes to add a new random component when the correlation coefficient between each data frame and its representation in terms of the current factors is lower than a threshold. This approach is insufficient here since the footprint of a new neuron in the whole FOV is typically too small to modify the correlation coefficient significantly.

We approach the problem by introducing a buffer that contains the last $l_b$ instances of the residual signal $\mathbf{r}_t = \mathbf{y}_t - A\mathbf{c}_t - \mathbf{b}\mathbf{f}_t$, where $l_b$ is a reasonably small number, e.g., $l_b = 100$. On this buffer, similarly to [26], we perform spatial smoothing with a Gaussian kernel with radius similar to the expected neuron radius, and then search for the point in space that explains the maximum variance. New candidate components $\mathbf{a}_{\text{new}}$, and $\mathbf{c}_{\text{new}}$ are estimated by performing a local rank-1 NMF of the residual matrix restricted to a fixed neighborhood around the point of maximal variance.

To limit false positives, the candidate component is screened for quality. First, to prevent noise overfitting, the shape $\mathbf{a}_{\text{new}}$ must be significantly correlated (e.g., $\theta_s \sim 0.8 - 0.9$) to the residual buffer averaged over time and restricted to the spatial extent of $\mathbf{a}_{\text{new}}$. Moreover, if $\mathbf{a}_{\text{new}}$ significantly overlaps with any of the existing components, then its temporal component $\mathbf{c}_{\text{new}}$ must not be highly correlated with the corresponding temporal components; otherwise we reject it as a possible duplicate of an existing component. Once a new component is accepted, $[A, \mathbf{b}]$, $[C; \mathbf{f}]$ are augmented with $\mathbf{a}_{\text{new}}$ and $\mathbf{c}_{\text{new}}$ respectively, and the sufficient statistics are updated as follows:

$$W_t = \left[ W_t, \frac{1}{t} Y_{\text{buf}} \mathbf{c}_{\text{new}}^\top \right], \qquad M_t = \frac{1}{t} \left[ \begin{array}{cc} t M_t & \tilde{C}_{\text{buf}} \mathbf{c}_{\text{new}}^\top \\ \mathbf{c}_{\text{new}} \tilde{C}_{\text{buf}}^\top & \|\mathbf{c}_{\text{new}}\|^2 \end{array} \right], \tag{5}$$

where $Y_{\text{buf}}, \tilde{C}_{\text{buf}}$ denote the matrices $Y, [C; \mathbf{f}]$, restricted to the last $l_b$ frames that the buffer stores. This process is repeated until no new components are accepted, at which point the next frame is read and processed. The whole online procedure is described in Algorithm 1; the supplement includes pseudocode description of all the referenced routines.

**Initialization:** To initialize our algorithm we use the CNMF algorithm on a short initial batch of data of length $T_b$, (e.g., $T_b = 1000$). The sufficient statistics are initialized from the components that the offline algorithm finds according to (3). To ensure that new components are also initialized in the darker parts of the FOV, each data frame is normalized with the (running) mean for every pixel, during both the offline and the online phases.

**Algorithmic Speedups:** Several algorithmic and computational schemes are employed to boost the speed of the algorithm and make it applicable to real-time large-scale experiments. In [19] block coordinate descent is used to update the factors $A$, warm started at the value from the previous iteration. The same trick is used here not only for $A$, but also for $C$, since the calcium traces are continuous and typically change slowly. Moreover, the temporal traces of components that do not spatially overlap with each other can be updated simultaneously in vector form; we use a simple greedy scheme to partition the components into spatially non-overlapping groups.

Since neurons' shapes are not expected to change at a fast timescale, updating their values (i.e., recomputing $A$ and $\mathbf{b}$) is not required at every timepoint; in practice we update every $l_b$ timesteps.

**Algorithm 1** ONACID

**Require:** Data matrix $Y$, initial estimates $A, \mathbf{b}, C, \mathbf{f}, S$, current number of components $K$, current timestep $t'$, rest of parameters.

1: $W = Y[:, 1 : t']C^\top/t'$
2: $M = CC^\top/t'$                ▷ Initialize sufficient statistics
3: $\mathcal{G} = \textsc{DetermineGroups}([A, \mathbf{b}], K)$         ▷ Alg. S1-S2
4: $R_{\text{buf}} = [Y - [A, \mathbf{b}][C; \mathbf{f}]][:, t' - l_b + 1 : t']$    ▷ Initialize residual buffer
5: $t = t'$
6: **while** there is more data **do**
7:      $t \leftarrow t + 1$
8:      $\mathbf{y}_t \leftarrow \textsc{AlignFrame}(\mathbf{y}_t, \mathbf{bf}_{t-1})$                     ▷ [16]
9:      $[\mathbf{c}_t; \mathbf{f}_t] \leftarrow \textsc{UpdateTraces}([A, \mathbf{b}], [\mathbf{c}_{t-1}; \mathbf{f}_{t-1}], \mathbf{y}_t, \mathcal{G})$    ▷ Alg. S3
10:     $C, S \leftarrow \text{OASIS}(C, \gamma, s_{\min}, \lambda)$                   ▷ [11]
11:     $[A, \mathbf{b}], [C, \mathbf{f}], K, \mathcal{G}, R_{\text{buf}}, W, M \leftarrow$
12:         $\textsc{DetectNewComponents}([A, \mathbf{b}], [C, \mathbf{f}], K, \mathcal{G}, R_{\text{buf}}, \mathbf{y}_t, W, M)$   ▷ Alg. S4
13:     $R_{\text{buf}} \leftarrow [R_{\text{buf}}[:, 2 : l_b], \mathbf{y}_t - A\mathbf{c}_t - \mathbf{bf}_t]$     ▷ Update residual buffer
14:     **if** $\mod (t - t', l_b) = 0$ **then**         ▷ Update $W, M, [A, \mathbf{b}]$ every $l_b$ timesteps
15:         $W, M \leftarrow \textsc{UpdateSuffStatistics}(W, M, \mathbf{y}_t, [\mathbf{c}_t; \mathbf{f}_t])$     ▷ Equation (3)
16:         $[A, \mathbf{b}] \leftarrow \textsc{UpdateShapes}[W, M, [A, \mathbf{b}]]$              ▷ Alg. S5
17: **return** $A, \mathbf{b}, C, \mathbf{f}, S$

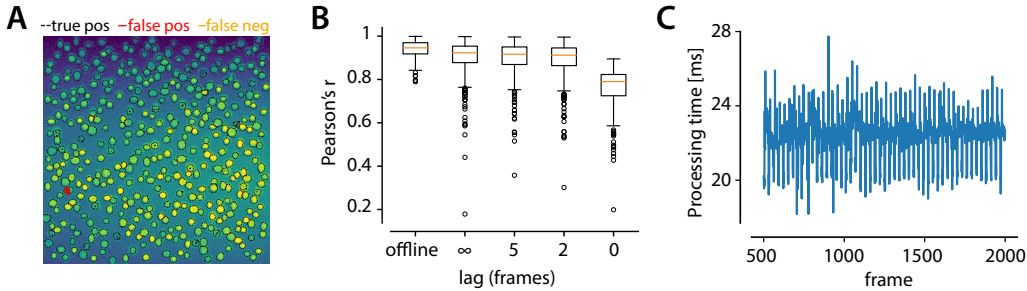

Figure 2: **Application to simulated data.** A) Detected and missed components. B) Tukey boxplot of spike train correlations with ground truth. Online deconvolution recovers spike trains well and the accuracy increases with the allowed lag in spike assignment. C) Processing time is less than 33 ms for all the frames.

Additionally, the sufficient statistics $W_t, M_t$ are only needed for updating the estimates of $[A, \mathbf{b}]$ so they can be updated only when required. Motion correction can be sped up by estimating the motion only on a small (active) contiguous part of the FOV. Finally, as shown in [10], spatial decimation can bring significant speed benefits without compromising the quality of the results.

**Software:** OnACID is implemented in Python and is available at `https://github.com/simonsfoundation/caiman` as part of the CaImAn package [13].

## 3 Results

**Benchmarking on simulated data:** To compare to ground truth spike trains, we simulated a 2000 frame dataset taken at 30Hz over a $256 \times 256$ pixel wide FOV containing 400 "donut" shaped neurons with Poisson spike trains (see supplement for details). OnACID was initialized on the first 500 frames. During initialization, 265 active sources were accurately detected (Fig. S2). After the full 2000 frames, the algorithm had detected and tracked all active sources, plus one false positive (Fig. 2A).

After detecting a neuron, we need to extract its spikes with a short time-lag, to enable interesting closed loop experiments. To quantify performance we measured the correlation of the inferred spike train with the ground truth (Fig. 2B). We varied the lag in the online estimator, i.e. the number of future samples observed before assigning a spike at time zero. Lags of 2-5 yield already similar

Table 1: OnACID significantly outperforms the offline CNMF approach. Benchmarking is against two independent manual annotations within the precision/recall (and their harmonic mean $F_1$ score) framework. For each row-column pair, the column dataset is regarded as the ground truth.

| $F_1$ (precision, recall) | Labeler 1 | Labeler 2 | CNMF |
|---|---|---|---|
| OnACID | 0.79 (0.87,0.72) | 0.78 (0.86,0.71) | 0.79 (0.83,0.75) |
| CNMF | 0.71 (0.74, 0.69) | 0.71 (0.75,0.68) | - |
| Labeler 2 | 0.89 (0.89,0.89) | - | - |

results as the solution with unrestricted lag. A further requirement for online closed-loop experiments is that the computational processing is fast enough. To balance the computational load over frames, we distributed here the shape update over the frames, while still updating each neuron every 30 frames on average. Because the shape update is the last step of the loop in Algorithm 1, we keep track of the time already spent in the iteration and increase or decrease the number of updated neurons accordingly. In this way the frame processing rate remained always higher than 30Hz (Fig. 2C).

**Application to** *in vivo* **2p mouse hippocampal data:** Next we considered a larger scale (90K frames, $480 \times 480$ pixels) real 2p calcium imaging dataset taken at 30Hz (i.e., 50 minute experiment). Motion artifacts were corrected prior to the analysis described below. The online algorithm was initialized on the first 1000 frames of the dataset using a Python implementation of the CNMF algorithm found in the CaImAn package [13]. During initialization 139 active sources were detected; by the end of all 90K frames, 727 active sources had been detected and tracked (5 of which were discarded due to their small size).

**Benchmarking against offline processing and manual annotations:** We collected manual annotations from two independent labelers who were instructed to find round or donut shaped neurons of similar size using the ImageJ Cell Magic Wand tool [31] given i) a movie obtained by removing a running 20th percentile (as a crude background approximation) and downsampling in time by a factor of 10, and ii) the max-correlation image. The goal of this pre-processing was to suppress silent and promote active cells. The labelers found respectively 872 and 880 ROIs. We also compared with the CNMF algorithm applied to the whole dataset which found 904 sources (805 after filtering for size).

To quantify performance we used a precision/recall framework similar to [2]. As a distance metric between two cells we used the Jaccard distance, and the pairing between different annotations was computed using the Hungarian algorithm, where matches with distance $> 0.7$ were discarded[4]. Table. 1 summarizes the results within the precision/recall framework. The online algorithm not only matches but outperforms the offline approach of CNMF, reaching high performance values ($F_1 = 0.79$ and $0.78$ against the two manual annotations, as opposed to $0.71$ against both annotations for CNMF). The two annotations matched closely with each other ($F_1 = 0.89$), indicating high reliability, whereas OnACID vs CNMF also produced a high score ($F_1 = 0.79$), suggesting significant overlap in the mismatches between the two algorithms against manual annotations.

Fig. 3 offers a more detailed view, where contour plots of the detected components are superimposed on the max-correlation image for the online (Fig. 3A) and offline (Fig. 3B) algorithms (white) and the annotations of Labeler 1 (red) restricted to a $200 \times 200$ pixel part of the FOV. Annotations of matches and mismatches between the online algorithm and the two labelers, as well as between the two labelers in the entire FOV are shown in Figs. S3-S8. For the automated procedures binary masks and contour plots were constructed by thresholding the spatial footprint of each component at a level equal to 0.2 times its maximum value. A close inspection at the matches between the online algorithm and the manual annotation (Fig. 3A-left) indicates that neurons with a strong footprint in the max-correlation image (indicating calcium transients with high amplitude compared to noise and background/neuropil activity) are reliably detected, despite the high neuron density and level of overlap. On the other hand, mismatches (Fig. 3B-left) can sometimes be attributed to shape mismatches, manually selected components with no signature in the max-correlation image (indicating faint or possibly unclear activity) that are not detected by the online algorithm (false negatives), or small partially visible processes detected by OnACID but ignored by the labelers ("false" positives).

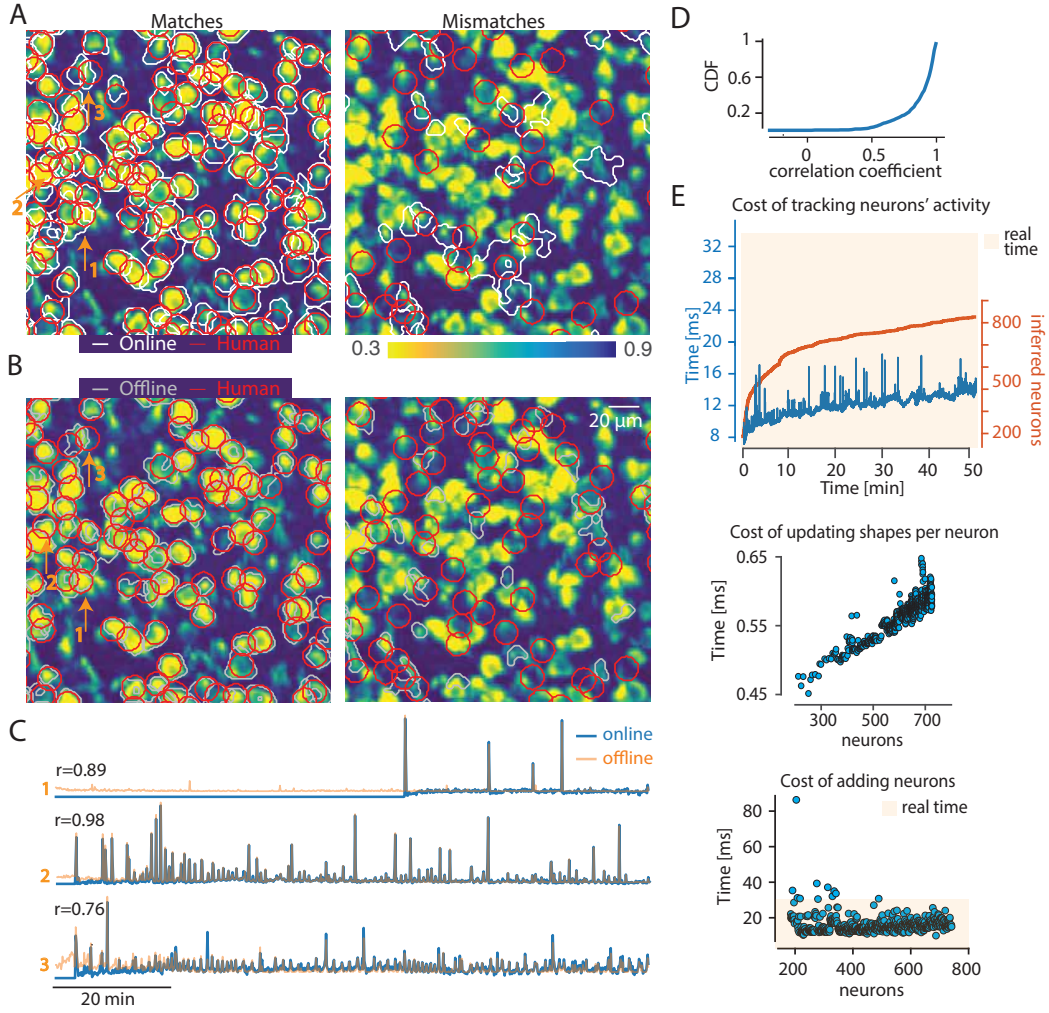

Figure 3: **Application to an *in vivo* 50min long hippocampal dataset and comparison against an offline approach and manual annotation.** A-left) Matched inferred locations between the online algorithm (white) and the manual annotation of Labeler 1 (red), superimposed on the max-correlation image. A-right) False positive (white) and false negative (red) mismatches between the online algorithm and a manual annotation. B) Same for the offline CNMF algorithm (grey) against the same manual annotation (red). The online approach outperforms the CNMF algorithm in the precision/recall framework ($F_1$ score 0.77 vs 0.71). The images are restricted to a $200 \times 200$ pixel part of the FOV. Matches and non-matches for the whole FOV are shown in the supplement. C) Examples of inferred sources and their traces from the two algorithms and corresponding annotation for three indentified neurons (also shown with orange arrows in panels A,B left). The algorithm is capable of identifying new neurons once they become active, and track their activity similarly to offline approaches. D) Empirical CDF of correlation coefficients between the matched traces between the online and the offline approaches over the entire 50 minute traces. The majority of the correlation coefficients has very high values suggesting that the online algorithm accurately tracks the neural activity across time (see also correlation coefficients for the three examples shown in panel C). E) Timing of the online process. Top: Time required per frame when no shapes are updated and no neurons are updated (top). The algorithms is always faster than real time in tracking neurons and scales mildly with the number of neurons. Time required to update shapes per neuron (middle), and add new neurons (bottom) as a function of the number of neurons. Adding neurons is slower but occurs only sporadically affecting only mildly the required processing time (see text for details).

Fig. 3C shows examples of the traces from three selected neurons. OnACID can detect and track neurons with very sparse spiking over the course of the entire 50 minute experiment (Fig. 3C-top), and produce traces that are highly correlated with their offline counterparts. To examine the quality of the inferred traces (where ground truth collection at such scale is both very strenuous and severely impeded by the presence of background signals and neuropil activity), we compared the traces between the online algorithm and the CNMF approach on matched pairs of components. Fig. 3D shows the empirical cumulative distribution function (CDF) of the correlation coefficients from this comparison. The majority of the coefficients attain values close to 1, suggesting that the online algorithm can detect new neurons once they become active and then reliably track their activity.

**OnACID is faster than real time on average:** In addition to being more accurate, OnACID is also considerably faster as it required $\sim$27 minutes, i.e., $\sim 2\times$ faster than real time on average, to analyze the full dataset (2 minutes for initialization and 25 for the online processing) as opposed to $\sim$1.5 hours for the offline approach and $\sim$10 hours for each of the annotators (who only select ROIs). Fig. 3E illustrates the time consumption of the various steps. In the majority of the frames where no spatial shapes are being updated and no new neurons are being incorporated, OnACID processing speed exceeds the data rate of 30Hz (Fig. 3E-top), and this processing time scales only mildly with the inclusion of new neurons. The cost of updating shapes and sufficient statistics per neuron is also very low ($<$ 1ms), and only scales mildly with the number of existing neurons (Fig. 3E-middle). As argued before this cost can be distributed among all the frames while maintaining faster than real time processing rates. The expensive step appears when detecting and including one or possibly more new neurons in the algorithm (Fig. 3E-bottom). Although this occurs only sporadically, several speedups can be potentially employed here to achieve beyond real time at *every* frame (see also Discussion section), which would facilitate zero-lag closed-loop experiments.

**Application to** *in vivo* **2p mouse parietal cortex data:** As a second application to 2p data we used a 116,000 frame dataset, taken at 30Hz over a 512$\times$512 FOV (64min long). The first 3000 frames were used for initialization during which the CNMF algorithm found 442 neurons, before switching to OnACID, which by the end of the experiment found a total of 752 neurons (734 after filtering for size). Compared to two independent manual annotations of 928 and 875 ROIs respectively, OnACID achieved $F_1 = 0.76, 0.79$ significantly outperforming CNMF ($F_1 = 0.65, 0.66$ respectively). The matches and mismatches between OnACID and Labeler 1 on a 200$\times$200 pixel part of the FOV are shown in Fig. 4A. Full FOV pairings as well as precision/recall metrics are given in Table 2.

Table 2: Comparison of performance of OnACID and the CNMF algorithm using the precision/recall framework for the parietal cortex 116000 frames dataset. For each row-column pair, the column dataset is regarded as ground truth. The numbers in the parentheses are the precision and recall, respectively, preceded by their harmonic mean ($F_1$ score). OnACID significantly outperforms the offline CNMF approach.

| $F_1$ (precision, recall) | Labeler 1 | Labeler 2 | CNMF |
|---|---|---|---|
| OnACID | 0.76 (0.86,0.68) | 0.79 (0.86,0.72) | 0.65 (0.55,0.82) |
| CNMF | 0.65 (0.70, 0.60) | 0.66 (0.74,0.59) | - |
| Labeler 2 | 0.89 (0.86,0.91) | - | - |

For this dataset, rigid motion correction was also performed according to the simple method of aligning each frame to the denoised (and registered) background from the previous frame. Fig. 4B shows that this approach produced strikingly similar results to an offline template based, rigid motion correction method [25]. The difference in the displacements produced by the two methods was less than 1 pixel for all 116,000 frames with standard deviations 0.11 and 0.12 pixel for the $x$ and $y$ directions, respectively. In terms of timing, OnACID processed the dataset in 48 minutes, again faster than real time on average. This also includes the time needed for motion correction, which on average took 5ms per frame (a bit less than 10 minutes in total).

## 4 Discussion - Future Work

Although at first striking, the superior performance of OnACID compared to offline CNMF, for the datasets presented in this work, can be attributed to several factors. Calcium transient events are localized both in space (spatial footprint of a neuron), and in time (typically 0.3-1s for genetic indica-

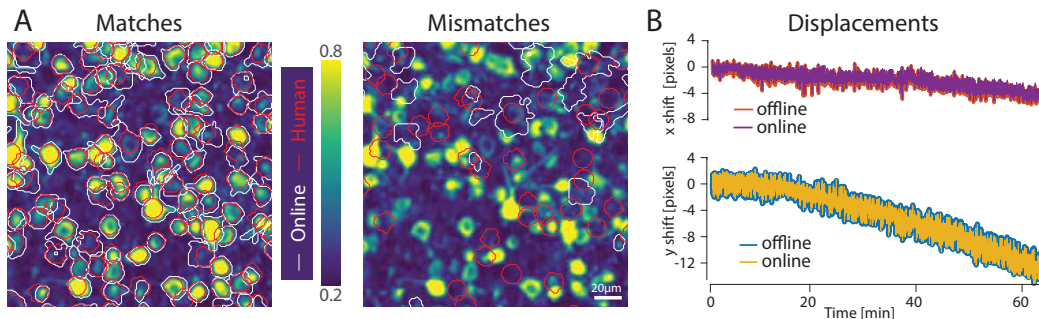

Figure 4: **Application to an *in vivo* 64min long parietal cortex dataset.** A-left) Matched inferred locations between the online algorithm (white) and the manual annotation of Labeler 1 (red). A-right) False positive (white) and false negative (red) mismatches between the online algorithm and a manual annotation. B) Displacement vectors estimated by OnACID during motion registration compared to a template based algorithm. OnACID estimates the same motion vectors at a sub-pixel resolution (see text for more details).

tors). By looking at a short rolling buffer OnACID is able to more robustly detect activity compared to offline approaches that look at all the data simultaneously. Moreover, OnACID searches for new activity in the residuals buffer that excludes the activity of already detected neurons, making it easier to detect new overlapping components. Finally, offline CNMF requires the a priori specification of the number of components, making it more prone to either false positive or false negative components.

For both the datasets presented above, the analysis was done using the same space correlation threshold $\theta_s = 0.9$. This strict choice leads to results with high precision and lower recall (see Tables 1 and 2). Results can be moderately improved by allowing a second pass of the data that can identify neurons that were initially not selected. Moreover, by relaxing the threshold the discrepancy between the precision and recall scores can be reduced, with only marginal modifications to the $F_1$ scores (data not shown).

Our current implementation performs all processing serially. In principle, significant speed gains can be obtained by performing computations not needed at each timestep (updating shapes and sufficient statistics) or occur only sporadically (incorporating a new neuron) in a parallel thread with shared memory. Moreover, different online dictionary learning algorithms that do not require the solution of an inverse problem at each timestep can potentially further speed up our framework [17].

For detecting centroids of new sources OnACID examines a static image obtained by computing the variance across time of the spatially smoother residual buffer. While this approach works very well in practice it effectively favors shapes looking similar to a pre-defined Gaussian blob (when spatially smoothed). Different approaches for detecting neurons in static images can be possibly used here, e.g., [22], [2], [29], [27].

Apart from facilitating closed-loop behavioral experiments and rapid general calcium imaging data analysis, our online pipeline can be potentially employed to future, optical-based, brain computer interfaces [6, 21] where high quality real-time processing is critical to their performance. These directions will be pursued in future work.

## Acknowledgments

We thank Sue Ann Koay, Jeff Gauthier and David Tank (Princeton University) for sharing their cortex and hippocampal data with us. We thank Lindsey Myers, Sonia Villani and Natalia Roumelioti for providing manual annotations. We thank Daniel Barabasi (Cold Spring Harbor Laboratory) for useful discussions. AG, DC, and EAP were internally funded by the Simons Foundation. Additional support was provided by SNSF P300P2_158428 (JF), and NIH BRAIN Initiative R01EB22913, DARPA N66001-15-C-4032, IARPA MICRONS D16PC00003 (LP).

## Footnotes

[1]These authors contributed equally to this work.

[2]To whom correspondence should be addressed.

[3]The correlation image (CI) at every pixel is equal to the average temporal correlation coefficient between that pixel and its neighbors [28] (8 neighbors were used for our analysis). The max-correlation image is obtained by computing the CI for each batch of 1000 frames, and then taking the maximum over all these images.

[4]Note that the Cell Magic Wand Tool by construction, tends to select circular ROI shapes whereas the results of the online algorithm do not pose restrictions on the shapes. As a result the computed Jaccard distances tend to be overestimated. This explains our choice of a seemingly high mismatch threshold.

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
