[Supplementary Material]

# Supplementary Material for "OnACID: Online Analysis of Calcium Imaging Data in Real Time"

Andrea Giovannucci, Johannes Friedrich, Matthew Kaufman, Anne K. Churchland,
Dmitri Chklovskii, Liam Paninski, and Eftychios A. Pnevmatikakis

## A Algorithmic description

Here we present in pseudocode the various steps of the online processing pipeline. For ease of exposition, some details and speedup tricks used in the actual implementation have been omitted.

Algorithms S1 and S2 describe the simple greedy procedure for partitioning the components into disjoint groups where the elements of each group do not overlap spatially with each other. This procedure is used for updating the traces of the neurons in vector form (Alg. S3) leading to substantial speed benefits. Algorithm S4 describes the procedure of detecting and screening possible new components. Finally, Algorithm S5 describes the process of updating the shapes, similar to [3].

---

**Algorithm S1** DETERMINEGROUPS

---

**Require:** Spatial components matrix $A$, number of components $K$

 1: $\mathcal{G} = \emptyset$
 2: **for** $i = 1 \to K$ **do**
 3:     $\mathcal{G} \leftarrow \text{JOINGROUPS}(A[:, 1 : i - 1], \mathcal{G}, i - 1, \mathbf{a}_i)$
 4: **end for**
 5: **return** $\mathcal{G}$

---

---

**Algorithm S2** JOINGROUPS

---

**Require:** Spatial components matrix $A$, current groups $\mathcal{G}$, number of components $K$, new component $\mathbf{a}$

 1: $N_G = |\mathcal{G}|$                                               $\triangleright$ number of groups
 2: repeat $=$ **True**
 3: $g \leftarrow 1$
 4: **while** repeat **do**
 5:     **if** $g \leq N_G$ **then**
 6:        **if** $\mathbf{a}^\top \mathbf{a}_l = 0, \forall l \in G_j$ **then**               $\triangleright$ Test for overlap with current group
 7:           $G_g \leftarrow G_g \cup \{K + 1\}$
 8:           repeat $=$ **False**
 9:        **else**
10:           $g \leftarrow g + 1$
11:        **end if**
12:     **else**
13:        $N_G \leftarrow N_G + 1$
14:        $G_{N_G} = \{K + 1\}$                                  $\triangleright$ Create a new group
15:        $\mathcal{G} \leftarrow \{\mathcal{G}, G_{N_G}\}$                              $\triangleright$ Add to list of groups
16:        repeat $=$ **False**
17:     **end if**
18: **end while**
19: **return** $\mathcal{G}$

---

**Algorithm S3** UPDATETRACES

---

**Require:** Spatial footprints matrix $\tilde{A} = [A, \mathbf{b}]$, current value of temporal traces $\tilde{\mathbf{c}} = [\mathbf{c}; f]$, current data frame $\mathbf{y}$, groups $\mathcal{G}$, tolerance level $\varepsilon$.

1: $\mathbf{u} = \tilde{A}^\top \mathbf{y}$
2: $V = \tilde{A}^\top \tilde{A}$
3: $\mathbf{V} = \mathrm{diag}\{V\}$
4: $\tilde{c}_{old} \leftarrow 0$
5: **while** $\|\tilde{c} - \tilde{c}_{old}\| \geq \varepsilon \|\tilde{c}_{old}\|$ **do**
6:      $\tilde{c}_{old} \leftarrow \tilde{c}$
7:      **for** $i = 1 \rightarrow |\mathcal{G}|$ **do**
8:          $\tilde{c}[G_i] = \max\left( \tilde{c}[G_i] + \frac{\mathbf{u}[G_i] - V[G_i,:]\tilde{c}}{\mathbf{v}[G_i]}, 0 \right)$                ▷ (Division is pointwise)
9:      **end for**
10: **end while**
11: **return** $\tilde{c}$

---

**Algorithm S4** DETECTNEWCOMPONENTS

---

**Require:** Spatial footprints matrix $[A, \mathbf{b}]$, temporal traces matrix $[C; \mathbf{f}]$, current number of components $K$, current state of groups $\mathcal{G}$, current residual buffer $R_{\mathrm{buf}}$, current data frame $\mathbf{y}$, sufficient statistics $W, M$. Parameters: radius of Gaussian kernel $\tau$, threshold for correlation in space $\theta_s$, threshold for correlation in time $r_t$.

1: repeat = **True**
2: $R_{\mathrm{buf}} \leftarrow [R_{\mathrm{buf}}[:, 1 : l_b - 1], \mathbf{y} - [A, \mathbf{b}][C; \mathbf{f}][:, \mathrm{end}]]$          ▷ Update residual buffer
3: $M_d = \mathrm{MEDIAN}(R_{\mathrm{buf}})$
4: $R_{\mathrm{buf}} \leftarrow R_{\mathrm{buf}} - M_d$          ▷ Subtract median along time for every pixel
5: $V \leftarrow \mathrm{FILTER}(R_{\mathrm{buf}}, \mathrm{GAUSSIANKERNEL}(\tau))$          ▷ Filter residual in space
6: $E \leftarrow \sum_i V[:, i].^2$          ▷ Compute energy value for each pixel
7: **while** repeat **do**
8:      $(i_x, i_y) = \arg\max E$          ▷ Find the point of maximum variance
9:      $N_{(i_x, i_y)} = \{(x, y) : |x - i_x| \leq \tau, |y - i_y| \leq \tau\}$          ▷ Define a neighborhood around $(i_x, i_y)$
10:      $[\mathbf{a}_{\mathrm{new}}, \mathbf{c}_{\mathrm{new}}] = \mathrm{NNMF}(R_{\mathrm{buf}}[N_{(i_x, i_y)}, :], 1)$          ▷ Perform a local rank-1 NMF
11:      $r = \mathrm{CORR}(\mathbf{a}_{\mathrm{new}}, \mathrm{MEAN}(R_{\mathrm{buf}}))$          ▷ Compute correlation coefficient in space
12:      $o = \mathrm{Find}(\mathbf{a}_{\mathrm{new}}^\top A[N_{(i_x, i_y)}, :] > 0$          ▷ Find components that overlap
13:      **if** $\exists j \in o : \mathrm{CORR}(\mathbf{c}_{\mathrm{new}}, C[j, t - l_b + 1 : t]) > r_t$ **then**
14:          $r \leftarrow 0$          ▷ Detect possible duplicates and stop procedure
15:      **end if**
16:      **if** $r > \theta_s$ **then**          ▷ New component is accepted
17:          Zero-pad $\mathbf{a}_{\mathrm{new}}$ and $\mathbf{c}_{\mathrm{new}}$ to match dimensionality
18:          $K \leftarrow K + 1$
19:          $\mathcal{G} \leftarrow \mathrm{JOINGROUPS}(A, \mathcal{G}, \mathbf{a}_{\mathrm{new}})$
20:          $A \leftarrow [A, \mathbf{a}_{\mathrm{new}}]$
21:          $C \leftarrow [C; \mathbf{c}_{\mathrm{new}}]$
22:          $R_{\mathrm{buf}} \leftarrow R_{\mathrm{buf}} - \mathbf{a}_{\mathrm{new}}\mathbf{c}_{\mathrm{new}}$
23:          $V \leftarrow V - \mathbf{a}_{\mathrm{new}}^2 \|\mathbf{c}_{\mathrm{new}}\|^2$
24:          $W, M \leftarrow \mathrm{UPDATESUFFSTATISTICS}(W, M, \mathbf{y}_t, \mathbf{c}_{\mathrm{new}})$          ▷ Equation (5)
25:      **else**
26:          repeat = **False**
27:      **end if**
28: **end while**
29: **return** $[A, \mathbf{b}], [C, \mathbf{f}], K, \mathcal{G}, R_{\mathrm{buf}}, W, M$

**Algorithm S5** UPDATESHAPES

---

**Require:** Sufficient statistics $W, M$, current value of spatial footprints $\tilde{A} = [A, b]$, list of components to be updated $l$, maximum number of iterations iterations $\mathrm{m_{iter}}$

1: iter $\leftarrow 0$
2: **while** iter $<$ $\mathrm{m_{iter}}$ **do**
3:     **for** $i \in l$ **do**
4:         $\mathbf{p} = \mathrm{find}(\tilde{A}[:, i] > 0)$                 $\triangleright$ Find the pixels where component $i$ can be non-zero
5:         $\tilde{A}[\mathbf{p}, i] = \max\left(\tilde{A}[\mathbf{p}, i] + \dfrac{W[\mathbf{p}, i] - \tilde{A}[\mathbf{p}, :]M[:, i]}{M[i, i]}, 0\right)$
6:     **end for**
7:     iter $\leftarrow$ iter $+ 1$
8: **end while**
9: **return** $\tilde{A}$

---

# B   Dataset Details

**Parietal cortex dataset:** Data was obtained from the parietal cortex of a transgenic GCaMP6f-expressing mouse during a behavioral task. Field of view was approximately $500{\times}500$ $\mu m^2$ ($512{\times}512$ pixels) in size and at depth $125\mu m$ below the dura surface. Horizontal scans of the laser were performed using a resonant galvanometer, resulting in a frame acquisition rate of 30Hz. More details can be found in [2].

    **Hippocampal dataset:** Data was obtained from the hippocampus of a transgenic GP2.11 (Thy1-GCaMP3) mouse generated by the Janelia Farms GENIE Project (Jackson Labs, C57BL/6J-Tg(Thy1-GCaMP3)GP2.11Dkim/J). FOV was approximately $500{\times}500$ $\mu m^2$, of size $512 \times 512$ pixels, cropped to 483 $\times$ 492 pixels after rigid registration and removal of empty border lines. Horizontal scans of the laser were performed using a resonant galvanometer, resulting in a frame acquisition rate of 30Hz. More details can be found in [1].

# C   Supplementary Movie

**Evolution of the OnACID algorithm on toy simulated data:** Top. Raw movie (left). Denoised movie reconstructed from all components (middle). Noiseless ground truth (right). Bottom. Residual movie (left). Inferred (middle) and ground truth (right) spatial components. A $64{\times}64$ pixel FOV containing 35 artifical neurons was simulated for this example. Movie is truncated in time for space reasons.

# D   Simulation details

We generated a dataset of size $256{\times}256$ pixels and duration $T = 2000$ frames containing $N = 400$ neurons. The neural centers $\{\mathbf{c}\}_1^N$ were generated using a Halton sequence to cover the space uniform pseudo-randomly.

    The unnormalized neural shapes were modeled as the difference of two 2D-Gaussians.

$$\tilde{a}(\mathbf{x}) = \exp\left(-\frac{1}{2}(\mathbf{x} - \mathbf{c})^\top \mathrm{diag}\left(\sigma_x^{-1}, \sigma_y^{-1}\right)(\mathbf{x} - \mathbf{c}))\right) \tag{1}$$

$$- k \exp\left(-\frac{1}{2}(\mathbf{x} - \mathbf{c})^\top \mathrm{diag}\left((0.75\sigma_x)^{-1}, (0.75\sigma_y)^{-1}\right)(\mathbf{x} - \mathbf{c}))\right) \tag{2}$$

where $\mathbf{x}$ denotes the position of the considered pixel. To incorporate heterogeneity the standard-deviation $\sigma_x$ and $\sigma_y$ in $x$- and $y$-direction of the wider Gaussian was drawn i.i.d. uniform randomly from the interval $[2.5, 3.5]$. These values were multiplied by 0.75 to obtain the standard-deviation of the smaller subtracted Gaussian. The magnitude $k$ of the subtracted Gaussian was drawn i.i.d. uniform randomly from the interval $[0.2, 0.8]$ for each neuron.

A

B

Figure S1: Simulated data. Detected components in batch mode. A) Full data B) Short initial batch.

The spike train **s** of each neuron was drawn from a homogeneous Poisson process. The neural firing rate was 0.5 Hz and the frame rate 30 Hz. The calcium traces $C$ were obtained by convolving the spike trains $S$ with an exponentially decaying kernel with time constant 1 s.

The background $B$ was modeled as rank 1 term, where the temporal and spatial component were each drawn from a Kronecker Gaussian process with RBF kernel. The temporal length scale was 300 frames and the spatial length scale 50 pixels. Finally, the simulated raw data is the sum of background $B$ and neural contribution $AC$ corrupted by Gaussian noise, $Y \sim \mathcal{N}(B + AC, 0.2^2)$.

Figure S2: Simulated data. Traces of three neurons selected by time of detection. Upper traces show demixed (red), denoised (orange, green) and ground truth (blue) calcium fluorescence. Lower traces show deconvolved neural activity using the same coloring scheme.

# E  Detailed comparison between OnACID and manual annotations for the hippocampal 2-photon dataset

In the following pages, Figures S3-S8 show the detailed matches and mismatches between OnACID and the two manual annotations, as well as the two manual annotations against each other.

Online (white) vs Labeler 1 (red) matches

Figure S3: Matches between OnACID (white) and Labeler 1 (red).

Online (white) vs Labeler 1 (red) mis-matches

Figure S4: Mismatches between OnACID (white) and Labeler 1 (red).

Online (white) vs Labeler 2 (red) matches

Figure S5: Matches between OnACID (white) and Labeler 2 (red).

Online (white) vs Labeler 2 (red) mis-matches

Figure S6: Mismatches between OnACID (white) and Labeler 2 (red).

Labeler 1 (red) vs Labeler 2 (white) matches

Figure S7: Matches between Labeler 1 (red) and Labeler 2 (white). The two labelers annotated the dataset independently and have a high degree of matching ($F_1 = 0.89$). The contour shapes are also similar for both annotators, as expected from the labeling process using the ImageJ Cell Wand tool.

Labeler 1 (red) vs Labeler 2 (white) mis-matches

Figure S8: Mismatches between Labeler 1 (red) and Labeler 2 (white).

# F    Detailed comparison between OnACID and manual annotations for the parietal cortex 2-photon dataset

In the following pages, Figures S9-S14 show the detailed matches and mismatches between OnACID and the two manual annotations, as well as the two manual annotations against each other.

Online (white) vs Labeler 1 (red) matches

Figure S9: Matches between OnACID (white) and Labeler 1 (red).

Online (white) vs Labeler 1 (red) mis-matches

Figure S10: Mismatches between OnACID (white) and Labeler 1 (red).

Online (white) vs Labeler 2 (red) matches

Figure S11: Matches between OnACID (white) and Labeler 2 (red).

Figure S12: Mismatches between OnACID (white) and Labeler 2 (red).

Figure S13: Matches between Labeler 1 (red) and Labeler 2 (white).

Labeler 1 (red) vs Labeler 2 (white) mis-matches

Figure S14: Mismatches between Labeler 1 (red) and Labeler 2 (white).