[Reviews · NeurIPS 2017]

Reviewer 1



This paper proposes an online framework for analyzing calcium imaging data. This framework is built upon the popular and now widely used constrained non-negative matrix factorization (CNMF) method for cell segmentation and calcium time-series analysis (Pnevmatikakis, et al., 2016). While the existing CNMF approach is now being used by many labs across the country, I’ve talked to many neuroscientists that complain that this method cannot be applied to large datasets and thus its application has been limited. This work extends this method to a real-time decoding setting, making it an extremely useful contribution for the neuroscience community. The paper is well written and the results are compelling. My only concern is that the paper appears to combine multiple existing methods to achieve their result. Nonetheless, I think the performance evaluation is solid and an online extension of CNMF for calcium image data analysis and will likely be useful to the neuroscience community. Major comments: - There are many steps in the described method that rely on specific thresholds or parameters. It would be useful to understand the sensitivity of the method to these different hyperparameters. In particular, how does the threshold used to include new components affect performance? - Lines 40-50: This paragraph seems to wander from the point that you nicely set up before this paragraph (and continue afterwards). Not sure why you’re bringing up closed loop, except for saying that by doing this in real time you can do closed-loop experiments. I’m not sure the idea that you want to communicate here. - The performance evaluations do not really address the use of isotonic regression for deconvolution. In Figure 3C, the traces appear to be demixed calcium after segmentation and not the deconvolved spike trains. Many of the other comparisons focus on cell segmentation. Please comment on the use of deconvolution and how it possibly might help in cell segmentation. - The results show that the online method outperforms the original CNMF method. Can the authors comment on where the two differ? Minor comments: Figure 1: The contours cannot be seen in B. Perhaps a white background in B (and later in the results) can help to see the components and contours. C is hard to see and interpret initially as the traces overlap significantly.

Reviewer 2



The authors present an online analysis pipeline for analyzing calcium imaging data, including motion artifact removal, source extraction, and activity denoising and deconvolution. The authors apply their technique to two 2-photon calcium imaging datasets in mouse. The presented work is of high quality. The writing is clear and does a great job of explaining the method as well as how it relates to previous work. The results are compelling, and the authors compare to a number of benchmarks, including human annotation. I encourage the authors to release source code for their analysis pipeline (I did not see a reference to the source code in the paper). Minor comments ----------------- - I was confused by the Methods section on motion correction. A few different methods are proposed: which is the one actually used in the paper? Are the others proposed as possible extensions/alternatives? - Unless I am mistaken, the claim that "OnACID is faster than real time on average" depends on the details of the dataset, namely the spatial and temporal resolution of the raw data? Perhaps this can be clarified in the text. - fig 3E: change the y-labels from "Time [ms]" to "Time per frame [ms]" (again, this depends on the spatial resolution of the frame?) - line 137: are the units for T_b in frames or seconds?

Reviewer 3



This paper describes a framework for online motion correction and signal extraction for calcium imaging data. This work combines ideas from several previous studies, and introduces a few innovations as well. The most pressing concern here is that there isn't a core new idea beyond the CMNF and OASIS algorithms. What is new here is the method for adding a new component, the initialization method, and the manually labeled datasets. This is likely quite a useful tool for biologists but the existence of a mathematical, computational or scientific advance is not so clear. Additionally, it's not so clear why we should trust the manual annotations. Ideally neurons would be identified by an additional unambiguous label (mcherry, DAPI, etc.). This, together with the relative similarity of CMNF and the current method call into question how much of an advance has been made here. The in vivo datasets need to be more clearly described. For example, Which calcium indicator was used? What happens if and when the focal plane drifts slowly along the z axis? It's also frequently mentioned that this is an online algorithm, but it's not clear what sort of latency can actually be achieved with the current implementation, or how it could be linked up with existing microscope software. If this is only possible in principle, the paper should stand better on its conceptual or computational merits. The spatial profiles extracted for individual neurons should be also shown.